# Metastatic Cutaneous Squamous Cell Cancer with Peritoneal Carcinomatosis: A Case Report

**Nikolaos Mitsimponas [1,*] and Anastasios Grivas [2]**

[1]  First Oncological Clinic, HYGEIA Hospital, Erythrou Stavrou 4 and Kifisias Avenue, Marousi, 15123 Athens, Greece

[2]  Second Oncological Clinic, Agios Savvas Hospital, Alexandras Avenue 171, 11522 Athens, Greece; tssgrvs@gmail.com

*  Correspondence: nikosmitsimponas@gmail.com; Tel.: +30-694-629-2127

**Abstract:** Peritoneal involvement as a metastatic site of squamous cell skin cancer is exceptionally rare. The current work analyzes a 52-year old male with high-risk cutaneous squamous cell nose carcinoma (cSCC) that was initially treated with surgery and platinum-based concurrent chemoradiation. Five years later, he presented jaundice and hypercalcemia. Further imaging revealed diffused liver, peritoneal and paraaortic lymph node metastases without evidence of locoregional recurrence. The patient underwent liver biopsy, which confirmed the diagnosis. High-risk features for metastasizing can be considered the maximum clinical diameter, the anatomical subsite (localization of the primary tumor in the ear and retroauricular area, cheek and lip are considered to significantly increase the risk of distant metastasis), poor histological differentiation, perineural invasion and lesions with a thickness of more than 2.0 mm. Late relapse that involves only disseminated abdominal disease is very uncommon and may justify closer follow-up and more aggressive chemotherapy in high-risk patients.

**Keywords:** cutaneous squamous cell cancer; peritoneal carcinomatosis; metastasis; malignancy of skin

## 1. Introduction

Squamous cell carcinoma is a common malignancy and the most usual sites of presentation are the lung, head and neck, esophagus, cervix, skin and anal canal [1]. Cutaneous squamous cell carcinoma is the second most common malignancy of the skin. Although rarely metastatic, squamous cell skin cancer can lead to substantial local destruction and may involve extensive areas of soft tissue, cartilage and bones [2,3]. Cutaneous squamous cell nose carcinoma (SCC) rarely metastasizes and the risk of distant metastatic disease is estimated at 1.9–2.6% [4]. High-risk features for metastasizing can be considered the maximum clinical diameter, the anatomical subsite (localization of the primary tumor in the ear and retroauricular area, cheek and lip are considered to increase significantly the risk of distant metastasis), poor histological differentiation, perineural invasion and lesions with a thickness of more than 2.0 mm. What makes the current work even more significant is the fact that it reports the first case in the international medical literature of peritoneal carcinomatosis as a metastatic site in a patient with cutaneous squamous cell cancer.

## 2. Case Presentation Section

A 52-year-old man presented in September 2017 with jaundice and hypercalcemia. He had a medical history of cSCC of the nose five years earlier. He had undergone wide local surgical excision without regional lymph node dissection, given that imaging studies were negative for lymph node

involvement. The primary tumor involved the mask area of the nose, it was 3.7 cm in diameter and had a Breslow depth of 1.6 cm extending to the nasal cartilage. Furthermore, it was poorly differentiated with an infiltrative growth pattern (desmoplastic). The pathology report also revealed positive surgical margins. The disease was staged as pT3cN0cM0 with high-risk features and the patient received platinum-based concurrent chemoradiation. He continued on his follow-up schedule. Five years after the first diagnosis our patient presented to our clinic with jaundice and hypercalcemia. Further imaging revealed diffuse liver, peritoneal and paraaortic lymph node metastases without evidence of locoregional recurrence. After a thorough workup the patient underwent a liver biopsy that confirmed the metastases from the known cSCC. He was administered a combination of cisplatin and fluorouracil chemotherapy without clinically meaningful response. Unfortunately, he passed away two months later due to liver metastatic disease.

## 3. Discussion

Cutaneous squamous cell carcinoma constitutes up to 20% of non-melanoma skin cancers. A number of risk factors are associated with the development of cSCC. The most recognized environmental carcinogen is the sunlight. Besides that, cSCC is also known to develop in association with scars or chronic wounds (Marjolin's ulcer) [5–8]. Actinic keratosis and Bowen's disease, if left untreated, can progress to invasive cSCC [9]. Other less common risk factors encompass prior therapy with immunosuppressive agents and immunosuppressed populations such as transplant recipients [10,11]. Certain genetic syndromes such as albinism and xeroderma pigmentosum and certain settings of immunosuppression (lymphoma, chronic lymphocytic leukemia and HIV) predispose affected individuals to cSCC.

Cutaneous squamous cell carcinomas that develop in the head and neck area or the genitalia, mucosal surfaces and ears are at greater risk of recurrence or metastasis than those that develop on the trunk and extremities. Pathologic risk factors for squamous cell skin cancer include the degree of differentiation, histology, depth of the lesion, and perineural, lymphatic and vascular involvement. Brougham et al. have proven that the maximum clinical diameter, the anatomical subsite (localization of the primary tumor in the ear and retroauricular area, cheek and lip are considered to significantly increase the risk of distant metastasis), poor histological differentiation and perineural invasion can be considered as high-risk features for metastasizing [4]. In our case, the primary tumor was located in the mask area of the nose; it had deep invasion and was poorly differentiated with an infiltrative growth pattern. Furthermore, the surgical margins were positive according to the pathology report. All these features underlined our case as a high-risk patient for developing local recurrence or metastasis. Brantsch et al. reported a metastasis-probability of 0% for tumors up to 2.0 mm in thickness, 4% for tumors 2.1 mm to 6.0 mm in thickness and 16% for tumors with a thickness greater than 6.0 mm [12]. A large retrospective analysis and a very large meta-analysis ($n = 17,248$ people) have shown that the risk of recurrence and metastasis is significantly higher for lesions thicker than 2.0 mm [13–15]. Some studies have shown significantly higher risk of recurrence or metastasis for cSCC lesions with Clark levels IV–V.

Standard treatment for high-risk patients with local cSCC is excision achieving wider margins. Adjuvant radiotherapy is recommended in patients with negative margins after surgery and large nerve or perineural involvement. In our patient, the presence of positive surgical margins and the other high-risk features were consistent with proceeding with platinum-based concurrent chemoradiation. Although metastatic peritoneal involvement from primary SCC has been rarely reported to occur with hypopharyngeal [16], lung [17], bladder [18] and cervical [19] primary cancers, this is the first case reported in the international medical literature presenting a metastatic peritoneal involvement with primary cutaneous squamous cell cancer. It should be underlined that cutaneous SCC with distant metastasis is extremely rare. Nevertheless, the most common metastatic sites of squamous cell skin cancer are lymph node (4.3%), liver (1.1%), lung (0.2%), bone (0.2%), brain (0.2%), subcutaneous tissue (0.2%), mediastinum (0.2%) [20]. Cisplatin, either as a single agent or combined with 5- Fluorouracil

(5-FU) or vindesine, has occasionally produced acceptable responses in patients with metastatic cSCC. In the current case, the patient received a combination of cisplatin and fluorouracil without response. Currently, tumoral cytoreduction followed by hypertermic intraperitoneal chemotherapy (HIPEC) is considered the treatment of choice for patients with Peritoneal carcinomatosis (PC). Unfortunately, although the HIPEC procedure can provide drastic improvements to life expectancy, skepticism exists regarding this approach, partly due to its high toxicity. Moreover, there is no benefit to using HIPEC without complete cytoreductive surgery [21]. Another promising approach to treat late-stage SCC in patients who cannot undergo resective surgery seems to be targeted therapy. In particular, targeted-based methods to induce apoptosis are currently considered a promising method for SCC anticancer therapy [22]. Though uncommon, the presence of metastatic disease in cSCC may underline the need for systemic therapy in high-risk patients.

## 4. Conclusions

Peritoneal carcinomatosis as a metastatic site of squamous cell skin cancer is an extremely rare entity, taking into consideration that the metastasis-probability of squamous cell skin cancers is inherently very low. High risk features for metastasizing can be considered the maximum clinical diameter, the anatomical subsite (localization of the primary tumor in the ear and retroauricular area, cheek and lip are considered to significantly increase the risk of distant metastasis), poor histological differentiation, perineural invasion and lesions with a thickness of more than 2.0 mm. Though uncommon, the presence of metastatic disease in cSCC may underline the need for systemic therapy in high-risk patients.

**Funding:** This manuscript received no external funding.

**Acknowledgments:** We gratefully acknowledge the assistance and contribution of Karatzas Konstantinos.

**Conflicts of Interest:** The authors declare no conflict of interest.

## Abbreviations

| | |
|---|---|
| cSCC | Cutaneous squamous cell cancer |
| 5-FU | 5-Fluorouracil |
| HIV | Human immunodeficiency virus |

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
