# Peer review of "Metastatic Cutaneous Squamous Cell Cancer with Peritoneal Carcinomatosis: A Case Report"

_reports, doi:10.3390/reports2010008_

Round 1

Reviewer 1 Report

This is a well written and interesting manuscript regarding a case report “Metastatic cutaneous squamous cell cancer with peritoneal carcinomatosis: a case report”. However, there are some comments that could be considered as follow:

1. Are there any images of the two biopsies?

Author Response

No images are not available, because biopsy was performed in another hospital.

Reviewer 2 Report

This case report will improve our knowledge regarding metastatic SCC. Overall, context will be acceptable for the IJMS. However, authors need to address following points

1)    Some words are not presented consistently, for example, P2, L88-89, cutaneous SCC should be cSCC.

2)    Discussion section should be improved. Sentence (P2, L71-74) totally miss information regarding thickness whereas rate of metastasis-probability and thickness are discussed here. Rather, the sentence should appear on ‘Case Presentation Section’

3)    Exactly same sentences are given in both abstract and other section, e.g. ‘Five years later he presented with・・・’. These parts should be revised.

Author Response

Could you specify, which words are not consistently presented. cSCC is cutaneous squamous cell carcinoma, as it is defined in the abbreviations.

There is no missing information about thickness, as it is defined in L80-83.

There is only one sentence same, which describes that the patient came to our clinic five years late with jaundice and hypercalcemia. I have made a correction in the lines 51-52 and i have uploaded it. 

Reviewer 3 Report

This is an interesting case report describing peritoneal carcinomatosis (PC) due to cutaneous squamous cell carcinoma (cSCC). PC represents the final stage of several types of advanced tumours, such as ovarian, gastric, or colorectal cancer. PC as a metastatic SCC is extremely rare and the Authors emphasize that this is the first case of PC in a patient with cSCC. There are few revisions needed before this work is suitable for publication:

1.     Only minor language corrections should be necessary:

·       The abbreviation cSCC must be introduced in the main text (Page 1, Line 28). Moreover, to improve readability, the terms “cutaneous squamous cell carcinoma” or “cutaneous SCC” must be replaced with “cSCC” (Page 1, Line 31 and 37; Page 2, Line 55, 64, 66, 88, 89; Page 3, Line 97 and 98);

·       Page 1, Line 32 and 43; Page 2, Line 44, 90, 91: replace Commas points (e.g. 1,9-2,6%) with Decimal points (e.g. 1.9-2.6%);

·       Page 1, Line 41: “cSCC of the nose” should be rephrased as “nasal cSCC”;

·       Page 1, Line 34: delete “are considered to increase significantly the risk of distant metastasis”;

·       Page 2, Line 52: “he passed away” should be rephrased as “the patient succumbed”;

·       Page 2, Line 62: “settings of immunosuppression” should be rephrased as “immunodeficiency disorders”.

2.     I suggest extending Discussion section by adding some considerations related to treatment of PC:

Currently, tumoral cytoreduction followed by hypertermic intraperitoneal chemotherapy (HIPEC) is considered the treatment of choice for patients with PC. Unfortunately, although HIPEC procedure can provide drastic improvements to life expectancy, skepticism exists regarding this approach partly due to high toxicity. Moreover, there is no benefit to using HIPEC without complete cytoreductive surgery [1]. Another promising approach to treat late-stage SCC in patients who cannot undergo resective surgery seems to be targeted therapy. In particular, targeted-based methods to induce apoptosis are currently considered a promising method for SCC anticancer therapy [2].

[1] Elias, D., Quenet, F., and Goéré, D. (2012). Current Status and Future Directions in the Treatment of Peritoneal Dissemination from Colorectal Carcinoma. Surg Oncol Clin N Am 21(4):611-23.

[2] Santarelli, A., Mascitti, M., Lo Russo, L., Sartini, D., Troiano, G., Emanuelli, M., and Lo Muzio, L. (2018). Survivin-Based Treatment Strategies for Squamous Cell Carcinoma. Int J Mol Sci 19(4).

Author Response

I agree with your corrections. 

I find your additional proposals very interesting. I have adapted them to the end of the discussion and your references at the end of the references list. You can check it in the uploaded file
